# SQL and NoSQL Databases in the Context of Industry 4.0

**Vitor Furlan de Oliveira \*** , **Marcosiris Amorim de Oliveira Pessoa** , **Fabrício Junqueira** and **Paulo Eigi Miyagi**

Escola Politécnica, Universidade de São Paulo, Sao Paulo 05508-030, Brazil; marcosiris@usp.br (M.A.d.O.P.); fabri@usp.br (F.J.); pemiyagi@usp.br (P.E.M.)
\* Correspondence: vitor.furlan@usp.br; Tel.: +55-(11)-96033-7582

**Abstract:** The data-oriented paradigm has proven to be fundamental for the technological transformation process that characterizes Industry 4.0 (I4.0) so that big data and analytics is considered a technological pillar of this process. The goal of I4.0 is the implementation of the so-called Smart Factory, characterized by Intelligent Manufacturing Systems (IMS) that overcome traditional manufacturing systems in terms of efficiency, flexibility, level of integration, digitalization, and intelligence. The literature reports a series of system architecture proposals for IMS, which are primarily data driven. Many of these proposals treat data storage solutions as mere entities that support the architecture's functionalities. However, choosing which logical data model to use can significantly affect the performance of the IMS. This work identifies the advantages and disadvantages of relational (SQL) and non-relational (NoSQL) data models for I4.0, considering the nature of the data in this process. The characterization of data in the context of I4.0 is based on the five dimensions of big data and a standardized format for representing information of assets in the virtual world, the Asset Administration Shell. This work allows identifying appropriate transactional properties and logical data models according to the volume, variety, velocity, veracity, and value of the data. In this way, it is possible to describe the suitability of relational and NoSQL databases for different scenarios within I4.0.

**Keywords:** Industry 4.0; database; data models; big data and analytics; asset administration shell





## 1. Introduction

Industry 4.0 (I4.0) designates the technological transformation process in production systems, logistics, and business models observed since the last decade [1]. The integration of digital technologies has promoted changes in the development phase [2,3], flexibility of production [3,4], efficiency in the use of resources [5,6], and level of automation and digitalization of the organizations [7,8]. This new mode of production characterizes the so-called Intelligent Manufacturing Systems (IMS): more efficient, flexible, integrated, and digitized than the traditional manufacturing systems. In the context of I4.0, the companies where the IMS are present are referred to as Smart Factories [9–11].

Data emerged as a fundamental resource for the Smart Factory due to their characteristics such as low cost, apparent inexhaustibility, and the possibility of cost reduction and value creation [12]. The authors of [10] argue that Smart Factory status is achieved, among other factors, when artificial intelligence solutions use the data. The "smart products" resultant from IMS are objects capable of storing and making their data available to humans or machines [9]. Thus, the importance of the data-oriented paradigm in the context of I4.0 is clear [12].

New system architectures have been proposed to promote the integration of enabling digital technologies to use data for industrial innovations [13–15]. While there is concern about optimizing IMS architectures in many respects, the impact of databases on their performance is not always considered. It is possible to observe that, in many cases, databases are treated as mere entities that support the functions of architectures, even though they can significantly influence the performance of the IMS [16].

This paper proposes the identification of data models that better suit different scenarios in the context of I4.0. This phenomenon is characterized by its key enabling data-related technologies and methods so that a consistent description of the nature of the data in this context could be achieved. By identifying the advantages and limitations of relational and NoSQL data models for such data characteristics, it is possible to discuss the suitability of these models for different scenarios in the context of I4.0.

## 2. Materials and Methods

Presenting the context in which this paper is inserted is essential to justify the choice of materials and methods adopted in the work from which it derives. For this reason, this section begins with a contextualization of Industry 4.0. Understanding some of its main characteristics is important to justify the relevance of this paper to (i) demonstrate the gap that exists in system architectures for I4.0 concerning data storage solutions used by these systems and (ii) identify patterns, methods, and technologies whose relevance to I4.0 is such that, from them, it is possible to obtain a characterization of the data in this context. Thus, this information is combined to propose suggestions for data models in the context of I4.0.

Nowadays, there is a consensual understanding that manufacturing automation systems have been undergoing a continuous transformation of technological paradigms since the last decade [1]. Authors claim that these transformations, obtained from integrating a series of independent digital technologies and a certain degree of independence from each other, configure the Fourth Industrial Revolution [17]. Because of the global scale of these changes, several initiatives worldwide, such as the Plattform Industrie 4.0 (https://www.plattform-i40.de/PI40/Navigation/EN/Home/home.html accessed on 2 August 2021), the Industrial Internet Consortium (https://www.iiconsortium.org/ accessed on 2 August 2021), and the Standardization Council Industrie 4.0 (https://www.sci40.com/ accessed on 2 August 2021), seek to establish guidelines for this process of technological transformation. The need to have a guide (or multiple equivalent guides) for the technological transformation process associated with the Fourth Industrial Revolution is because, unlike the first three, the Fourth Industrial Revolution was identified as such already in its early stages. Thus, these initiatives become responsible for outlining the advancement of technological transformation in manufacturing, proposing a common understanding of the phenomenon, establishing standards, and so on.

Among the different technological aspects mentioned above, some are highlighted in this work and focus is given to the so-called I4.0, a term often used as a synonym for the Fourth Industrial Revolution. In Germany, the Plattform Industrie 4.0 was created, a consortium of various organizations, including industries, universities, and the German government, proposing to shape the digital transformation in manufacturing according to the precepts of I4.0. The meaning of the term "Industry 4.0" is the object of analysis by several researchers [18–20]. Instead of presenting a definition, the option is to describe the phenomenon in terms of its characteristics: I4.0 is characterized by Intelligent Manufacturing Systems (IMS) that quickly adapt to market demands and with effective interconnection between all entities involved in these processes. This phenomenon is the conception of the so-called Smart Factory, which aims at manufacturing based on intelligent services and processes [21].

The main materials used in this work were technical publications and academic works. Considering that I4.0 is the result of cooperation between academia, industry, and government organizations, it was impossible to use only literature review methods of academic publications. Characterizing a system's data is essential for choosing the database to be adopted in the architecture for this system. In this work, this characterization is made based on technologies, methods, and standards for data in I4.0. Other relevant features for database design are fundamentally application dependent and are beyond the scope of this paper which seeks to expand the coverage of its contributions. The following paragraphs

describe the materials and methods adopted to characterize data in different scenarios in the context of I4.0 and identify which data models are suitable for different scenarios.

Ensuring interoperability among systems is one of the requirements for implementing the Smart Factory [20,22,23]. For this purpose, the entities that establish guidelines for the advancement of I4.0 proposed a standard format for digitally representing and managing elements involved in carrying out productive activities—the Asset Administration Shell (AAS)—whose concept, structure, metamodel, and perspectives for implementation will be presented in Section 3.2. Current works propose the implementation and use of the Asset Administration Shell in system architectures that seek to use data for different purposes. However, it is noted that less attention is paid to the design of the database to be used in these architectures. To confirm this statement, a systematic review of the literature was carried. The adopted procedures were the following:

- The paper databases used in the search were: Web of Science, IEEEXplore, Science Direct, Scopus, and Google Scholar;
- The following search string was defined to find papers: "Asset Administration Shell" AND "Database";
- It was observed that, among the selected databases, the only one to return a considerable number of papers was Google Scholar, which included papers from the other databases and, therefore, was the only one used. The application of search string returned 139 papers;
- The following keywords were defined for ranking the papers: "AAS"; "Asset Administration Shell"; "Database"; "DBMS" (database management system); "Implement*"; and "Storage";
- Each occurrence of any keyword in the title of the paper assigned 5 points to it (the Google Scholar platform does not allow exporting the abstract or keywords of the article). For instance, the paper entitled "Toward Industry 4.0 Components: Insights into and Implementation of Asset Administration Shells" contains the keywords implementation and "Asset Administration Shell" so it scored 10 points;
- Papers with a score greater than or equal to 5 were classified as accepted, and their content was analyzed;
- It was researched which of the papers classified as accepted cited the implementation data model and/or DBMS used.

Considering that I4.0 is a process of technological transformation, important data-based digital technologies and methods were identified. Those have such importance for this process that a description of the nature of the data in the context of I4.0 can be obtained from them. In addition to academic works, technical publications such as working papers from key organizations and entities for I4.0 were also considered in this process.

## 3. Basic Concepts

This section presents a theoretical framework composed of essential basic concepts for the work. Database-related topics include relational and NoSQL data models, transactional properties, and theorems regarding these properties. Moreover, the Asset Administration Shell concept, an artifact developed to represent Industry 4.0 components in the digital world, is presented.

### 3.1. Relational and NoSQL Databases

A logical and coherent collection of data with an intrinsic meaning forms a database [24]. A database stores and ensures the persistence and integrity of data that represent assets, in addition to allowing these data to be made available to interested users. A database is created and maintained through a database management system (DBMS), a computer program that helps maintain and use data sets that compose the databases [25]. These programs have the following advantages: they enable efficient and concurrent access to data; ensure data integrity and security; protect against failures and unauthorized access;

support multiple views of data; and, finally, they guarantee independence, that is, the isolation between data and applications through data abstraction [25,26].

Data abstraction is provided through data models. A data model is a set of concepts used to describe the structure of a database [24]. The logical data model describes data in such a level of abstraction that hides some details of the physical storage, which allows the end-user of the data to understand them. At the same time, as they are not so far from the low level, these concepts can be used directly to implement a database in a computer system. DBMSs are usually characterized by the logical data models they implement and, for this reason, this work focuses on this level of data abstraction.

### 3.1.1. Relational Data Model

The relational data model was, for many years, the default choice for database implementation [27]. It uses the concept of "relation" in a mathematical sense to represent data. Instead of presenting a formal mathematical definition of the term, which can be found in [28], it is presented how a relationship is perceived. Relations can be seen as tables of values. These tables have columns not necessarily distinct that consist of "attributes" used to characterize an element to be represented by the relation. Each line (formally called "tuple") of this table has values for the attributes. For each column, the values present in every tuple belong to a single domain with a well-defined name, data type, and format. Besides, only atomic values (each value in the domain is indivisible) are allowed.

A schema defines the structure of a relational database. From a schema, tables, their attributes, and relationships between them can be described to be used through a DBMS to create a database. The vast majority of DBMSs that implement a relational model use a standard language to perform queries—the Structured Query Language (SQL); the relational model is commonly called the SQL model. The same extends to the DBMSs and databases that implement it.

### 3.1.2. ACID and BASE Transactional Properties and CAP Theorem

Relational DBMS grants four properties to transactions to maintain data through concurrent access and system failures. These properties are atomicity (A), consistency (C), isolation (I), and durability (D), so they are often referred to as ACID properties. A brief description of each of them based on [25,26] is presented:

- Atomicity: The transaction must be executed in its entirety or not to be executed. If during the transaction, any failure occurs that prevents the transaction from being completed, any changes that it has performed in the database must be undone;
- Consistency: If a transaction runs entirely from start to finish, without interference from other transactions, it should take the database from one consistent state to another. A consistent database state satisfies the constraints specified in the schema as well as any other database constraints that must be maintained;
- Isolation: The execution of a transaction must not be interfered with by any other transaction running at the same time;
- Durability: Changes applied to the database by a committed transaction must persist in the database. These changes must not be lost due to any failure.

A distributed database is defined as "a collection of several logically interrelated databases distributed over a network of computers" [29]. There are three crucial reasons pointed out in the literature for the use of distributed databases: (1) the increase in the volume of data [27], which requires the ability to scale horizontally, that is, to distribute the systems across several nodes—instead of vertically scaling the hardware, adding more computing resources to the same machine, which would be more expensive and limited; (2) the need to better reflect the distributed organizational structure of companies [30]; and (3) the inherently distributed nature of a range of applications, including industrial ones [30]. There are three ways to implement a distributed database system [26,27]. It is noteworthy that specific systems implement hybrid versions, combining different forms of partitioning [27].

- Single server: There is no distribution. The database runs on a single machine that handles all operations. It is an example of a centralized AAS implementation;
- Partitioning: Different pieces of data on different machines. Aggregate models are ideal here, as they form a natural partitioning unit, making certain users access, most of the time, the same server so there is no need to gather information from different servers, which increases performance compared to a single server implementation;
- Replication: Data can be replicated in a master–slave schema where the master processes updates and replicates this data to other nodes or in a peer-to-peer schema where all nodes can process updates and propagate them.

Three other properties are desirable for database systems: availability (A), partition tolerance (P), and consistency (C), which, in this case, has a slight difference from the concept presented before. The analysis of these properties is fundamental in distributed database systems. The CAP theorem correlates these three properties. According to it, these three properties, whose descriptions are presented here, cannot coexist simultaneously in a database system:

- Consistency (C): Ensuring that all nodes have identical copies of replicated data visible to applications. It is a little different from the consistency concept of ACID properties. In that case, consistency means not violating database restrictions. However, it can be considered that "having the same copy of a data replicated in all nodes that this data is replicated on" is a restriction, so the concepts start to resemble each other;
- Availability (A): Each write or read operation will be successfully processed (system available) or will fail (system unavailable). A "down" node is not said to be unavailable;
- Partition Tolerance (P): A partition tolerant system continues to operate if a network fails to connect nodes, resulting in one or more network partitions. In this case, nodes in a partition only communicate with each other.

According to the CAP theorem, only two of the three properties presented can be guaranteed simultaneously. It is worth noting that this choice is not binary, it is possible to relax specific properties so that it is possible to privilege others. However, ACID transactional properties make this flexibility unfeasible. As a kind of alternative, you can have a database that works basically all the time (basically available) and is not consistent all the time (flexible state), only when the writes are propagated to all nodes (eventually consistent) in a distributed system. Thus, the characteristics of this model, named BASE model, although not strictly defined, are presented as:

- Basically available (BA): the system must be available even if partial failures occur;
- Flexible State (S): the system may not have consistent data all the time;
- Eventually consistent (E): consistency will be achieved once all writes are propagated to all nodes.

### 3.1.3. NoSQL Data Models

NoSQL does not have a solid definition, but is possibly better understood as a movement that proposes non-relational database solutions that do not use the SQL language [27,31]. Thus, the term NoSQL (often interpreted as Not Only SQL), used in its technical sense, is applied to designate a family of DBMSs that have specific characteristics in common (at least for most DBMSs), the main one being the non-implementation of a relational data model [32]. These features can mean advantages or disadvantages for specific applications:

- They do not implement the relational data model: Self-description and the absence of a fixed schema allow for greater flexibility concerning the content stored in DBMS, being suitable for handling semi-structured data [26,32];
- They do not use the SQL language: The absence of a declarative query language, with a wide range of "features" that are sometimes unnecessary, requires more significant

effort for developers since the functions and operations have to be implemented through the language of programming [26];

- Absence of ACID transactions: Because aggregate-oriented data models generally do not guarantee ACID transactional properties, DBMSs that implement these models have greater efficiency in distributed systems [32,33]. Alternatively, these DBMSs use the BASE model of transactional properties;

- Horizontally scalable: The ability of NoSQL DBMS to scale out is linked to two main characteristics. (1) By not having ACID transactional properties (aggregate-oriented models), it allows for relaxing consistency, and thus balancing the consistency–latency trade-off in the way which is most suitable for the application, without giving up partition tolerance, as it was previously discussed. (2) The orientation to aggregates allows for a "natural" or intuitive data partitioning unit, as data from an aggregate are commonly accessed together and can be allocated on the same server, which makes the user of this data access, in the majority sometimes, the same server [27].

NoSQL DBMS are commonly differentiated based on how they store the data, that is, the data model employed for the storage. There are four main data models implemented in NoSQL DBMSs. The description of each of these models is presented here:

- Key-value: Key-value DBMSs are possibly the simplest NoSQL systems. These DBMSs store their data in a table without a rigid schema, where each line corresponds to a unique key and a set of self-described objects called value. These can take different formats, from the simplest as strings, passing through tables as in the relational model, reaching more elaborate formats such as JavaScript Object Notation (JSON) and eXtensible Markup Language (XML) documents. Thus, they can store structured, semi-structured, and unstructured data in one format (key, value). The key-value data model is often represented as a hash table. This data model is aggregate oriented, meaning that each value associated with a unique key can be understood as an aggregate of objects that can be retrieved in their entirety through the key. The content of these aggregates can be different for each key. The aggregate's opacity guarantees the possibility of storing any data in the aggregates; that is, the DBMS does not interpret the aggregate content, seeing it only as a set of bits that must always be associated with its unique key. This has the practical implication of generally not allowing partial retrievals on aggregate content. The operations implemented by key-value DBMSs are the insertion or update of a pair (key, value), the retrieval of a value from its key, or the deletion of a key;

- Documents: Document-oriented DBMSs are those in which data is stored in document format. They can be understood as a key-value DBMS in which the only allowed formats for the values are documents such as XML, JSON, or PDF. A fundamental difference between the document and key-value data models is that the former allows for partial aggregate retrievals as it stores self-described data format. In other words, aggregates are not opaque, they are not seen by the DBMS merely as a set of bits, and it is possible to define indexes on the contents of the aggregates that allow operations to be performed on specific items of this data set. As with the key-value data model, the content of each document does not follow a fixed schema. Document labels that guarantee the self-description of the data and enable partial recoveries also allow different keys to have documents with different content (attributes). Thus, it allows the storage of structured and semi-structured data. There are still DBMSs that allow the storage of unstructured data such as texts;

- Column family: In column family databases, data is stored similarly to key-value databases. However, the value can only be composed of a set of tables, each of which has a name (identifier) and forms a column family. In each of these tables, columns are self-described; they have a key (also called a qualifier) and its value, which is the data itself. Thus, a column family database is formed by a table without a rigid structure containing unique keys and a series of column family associated with each key in each row. Some considerations can be made about this model. The first is that keys do

not need to have the same column family. The second observation is that, for each key, each column family can only contain the columns of interest; that is, the column family does not need to be composed of the same columns for all the keys. The column family forms a data aggregate that is frequently accessed together and, because these columns contain their keys, the aggregates are not opaque to the DBMS, thus being possible to perform partial recoveries through the aggregates through the indexes of the columns;

- Graphs: In graph-oriented DBMSs, data is stored in a collection of nodes, which represent entities, and directed vertices, which represent relationships among these entities. The set of nodes and vertices form graphs in which the two elements that compose them can contain labels and attributes associated with them, which are the data itself. Regarding the characteristics of the data models presented so far, the flexibility in data representation due to the absence of a fixed schema is one of the few similarities between the graph data model and the other data models mentioned, as both vertices and nodes can contain attributes different from each other. Concerning differences, the graph model is not aggregate oriented, it usually has ACID transactional properties, it is best suited for single server (non-distribution) implementations, can represent small records with complex relationships to each other, and it is more efficient in identifying patterns. Unlike aggregate-oriented models, where partial recoveries can only be made on one aggregate at a time (when allowed), in the graph model they can be conducted for the graph as a whole.

### 3.2. Asset Administration Shell

Before presenting the Asset Administration Shell (AAS), it is necessary to introduce the concept of "asset". The IEC describes an asset as "a physical or logical object owned or held in custody by an organization, having a perceived or real value to the organization". Based on this definition, also adopted by the Plattform Industrie 4.0 [34], it is possible to recognize that an asset can be something physical (a machine, equipment, materials, products) or not (electronic documents, computer programs). Some less intuitive examples of assets are location, time, state of an asset, human beings, and relationships between assets [35]. In summary, the asset is everything that has value and importance for an organization.

It is known that I4.0 characterizes a digital transformation process. For this process to occur, the assets need to be digitized; their data must be taken to the virtual world [36,37]. To perform this mapping to ensure interoperability between systems and components [37] in IMS, the AAS was created. It corresponds to a standardized digital representation of the asset containing all its technical information and functionalities. The AAS provides a minimum, unique, and sufficient description of the asset in different perspectives relevant to each use case [34,38]. By standardizing the representation format and communication interfaces of assets in the digital world, AAS enables the exchange of information among I4.0 participants, ensuring interoperability between components [34,38]. In summary, the AAS corresponds to the virtual and standardized representation of an asset in the context of I4.0.

The combination of asset and AAS gives rise to Component I4.0 (I4.0C or I4.0 Component). The I4.0C combines the physical and real world, composed by the asset and its respective AAS. The combination of these two elements, with the second "involving" the first, allows services and functionalities to be offered inside (through AAS) and outside (through the asset) of the I4.0C network. These features and services are made possible by the unique identification and communication capability of an I4.0C. Here, it is worth noting that a single I4.0C can be associated with multiple assets depending on the considered granularity. In this way, such a structure can be replicated to different levels of granularity (for example, at different levels of hierarchy). The following subsections discuss details of the AAS structure, elements, metamodel, and implementation perspectives.

The elements that compose an AAS are divided into classes; each has its attributes used to describe the asset. Elements in the AAS can be understood as subclasses, which

have the same attributes as a specific superclass from which they are derived but also contain attributes that differentiate them from other elements of the same superclass. The first way to divide the element classes of an AAS is to designate them as Identifiable and Referable. Identifiable Elements have a globally unique identifier. Referable, in turn, has an identifier that is not globally unique, being unique only within the context (defined by an Identifiable) in which it finds itself. The element classes contained within the Identifiable and Referable superclasses are called subclasses and can also, in turn, be superclasses; that is, they can be composed of other subclasses. There is an inheritance relationship of attributes in this hierarchy between classes: subclasses inherit attributes from superclasses.

The Identifiable Elements class can present additional domain-specific (owner) identifiers. The "Asset Administration Shell" and "Asset" subclasses have already been described from the Identifiable class. The subclass "Conceptual Description" defines the standardized semantic description of certain elements. Finally, the subclass "Submodel" allows an asset to be represented in its different perspectives. Each Submodel can describe an asset from an electrical, mechanical, thermal, control, and other perspective.

The "Referable Elements" class has more subclasses than the "Identifiable Elements" class, so only some of them are presented here in more detail. The description of all subclasses can be found in [34]. Among the subclasses of Referable elements, the subclass "Submodel Element" stands out in this work, and it is composed of elements suitable for the description and differentiation of assets in perspective specified by the Submodel. This class of "Submodel Elements" can be understood as a superclass in which some of the main subclasses are "Submodel Element Collection"—a collection that can be composed of all other classes with the same hierarchical level—and "Data Element". Data Elements, in turn, form another superclass with one of the important AAS element subclasses, the "Property", described in more detail in the next paragraph.

The "Properties" class contains elements that allow representing the characteristics of an asset given a perspective defined by the Submodel in which they are found. These elements are standardized by the IEC 61360 and can be found in the IEC Common Data Dictionary (CDD, common data dictionary) or eCl@ss repositories [34,39,40]. In the IEC CDD repository, a property has, in addition to its value itself, some additional data such as code, version and revision, identifier, and definition. The free digital version of the IEC CDD provides examples of properties for some specific domains. The complete list of AAS element classes, including those that do not qualify as Identifiable or Referable, can be found in [34].

Once the structure and some of the main components of the AAS are presented, it is possible to illustrate its metamodel. Figure 1 illustrates this metamodel with the components that were presented through a UML class diagram. The representation of AAS in the diagram also contains an example for the content of the AAS elements to highlight that its strict, coherent structure can be composed of data in different formats to contemplate the heterogeneity of assets represented through this artifact. A generic servo motor was considered as the asset to be described by the AAS. In Figure 1, the acronyms SM, SMC, Prop, and CD stand for Submodel, SubmodelElementCollection, Property, and ConceptDescription, respectively.

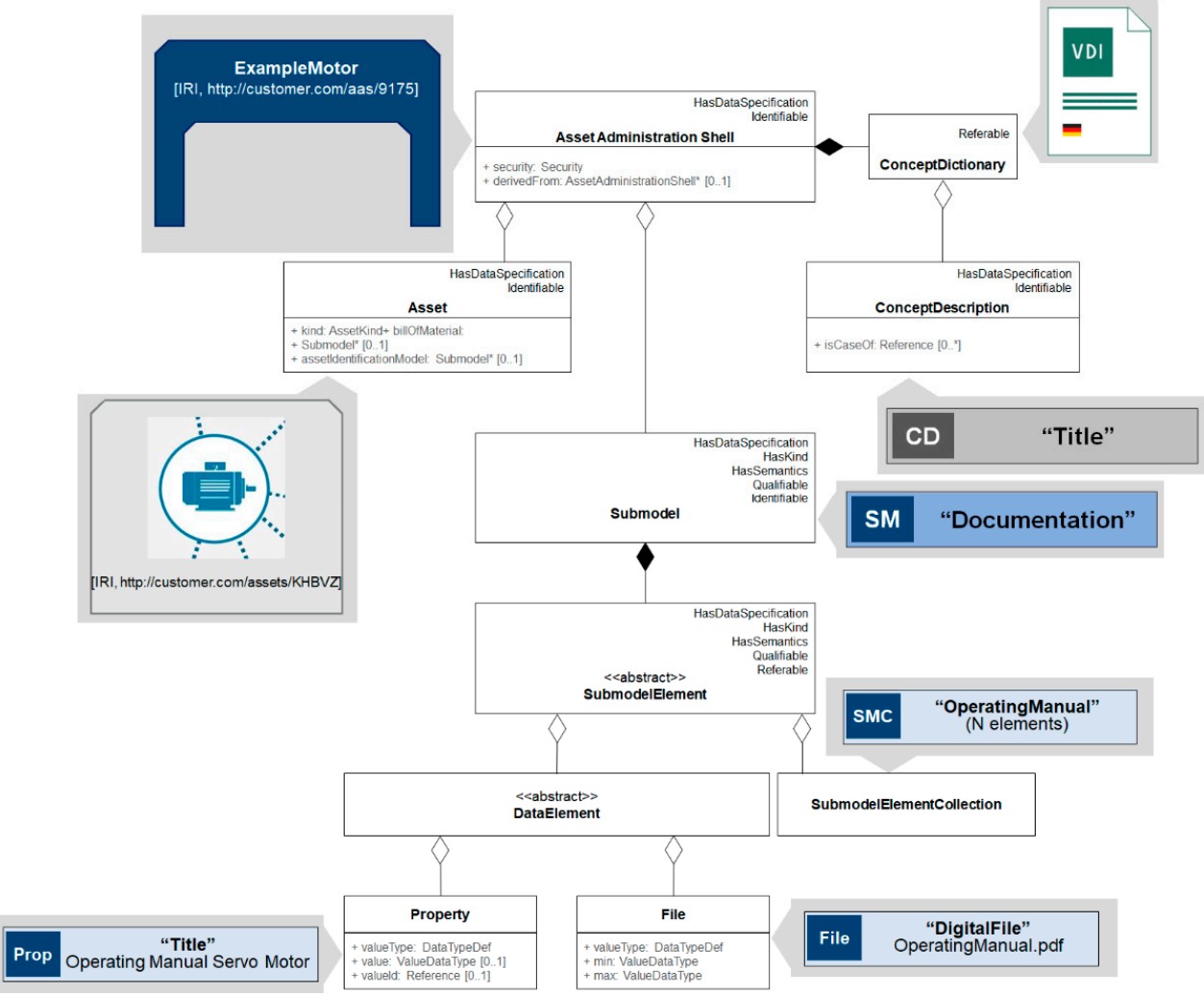

**Figure 1.** UML class diagram representing the metamodel of the AAS with main elements. Based on a generic servo motor as an asset, an example for the content of the AAS elements is also provided. Adapted from [34,41].

The standard solutions proposed for I4.0 need to be comprehensive enough not to limit the possibility of carrying out the Smart Factory in most different organizations. In this sense, no specific strategy for implementing the AAS is imposed by standardizing the digital format of representation and exchange of information. In [42], some of these possibilities are explored, taking into account different implementation perspectives. The different possibilities provide characteristics that can be advantages or disadvantages for specific applications. These characteristics include computational power, availability, performance and latency, security and reliability, maintenance, administration and management cost, failure identification, and recovery. Here, three perspectives of AAS implementation presented in [42] are described, along with the advantages and disadvantages.

The first issue to be discussed regarding AAS implementation is the computing platform. Three possibilities of implementation are presented, as illustrated in Figure 2. In the first one (Figure 2a), the AAS is embedded in the asset which, in turn, contains an execution environment for its digital representation. It is the case that an implementation based on an edge computing platform can be used as an example for such an implementation. In a second possibility (Figure 2b), the AAS can be physically separate from the asset but residing in the local IT infrastructure, connected to the asset through a local network. This case corresponds to a fog computing platform-based implementation. As a third possibility

(Figure 2c), the location of the AAS can be even further away from the asset in a cloud computing platform-based implementation. In this case, AAS and assets connect via external internet networks.

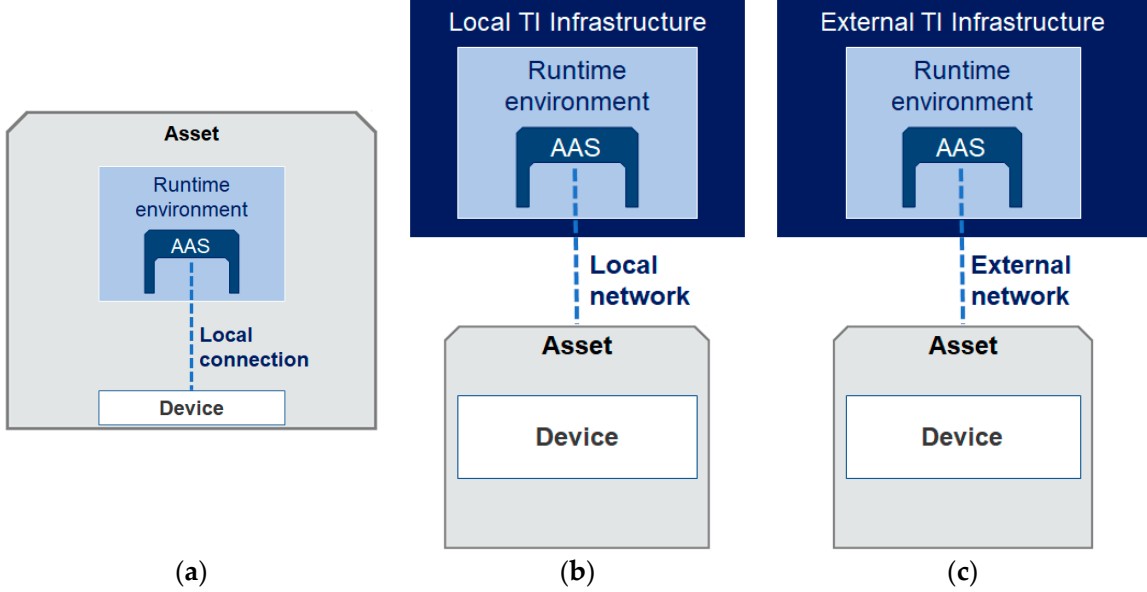

(**a**)  (**b**)  (**c**)

**Figure 2.** Computing platform perspective for implementation of the AAS: (**a**) edge, (**b**) fog, (**c**) cloud. Adapted from [42] with permission from the Federal Ministry for Economic Affairs and Climate Action. Scharnhorststraße 34–37, 10115 Berlin.

The second implementation perspective concerns the scalability of AAS. In a simplified way, scalability is related to the possibility of distributing data storage and processing across multiple nodes of a network. In this subsection, three possibilities for AAS distribution are considered, as illustrated in Figure 3: Figure 3a centralized, in which all information and services are allocated in a single node; Figure 3b loosely coupled distributed, where different nodes store information for the same asset (same identifier) and can be accessed individually; and Figure 3c distributed with aggregator node, which differs from the previous implementation by including an aggregator node, which gathers information from the nodes on which the AAS is distributed, forming a single data access point.

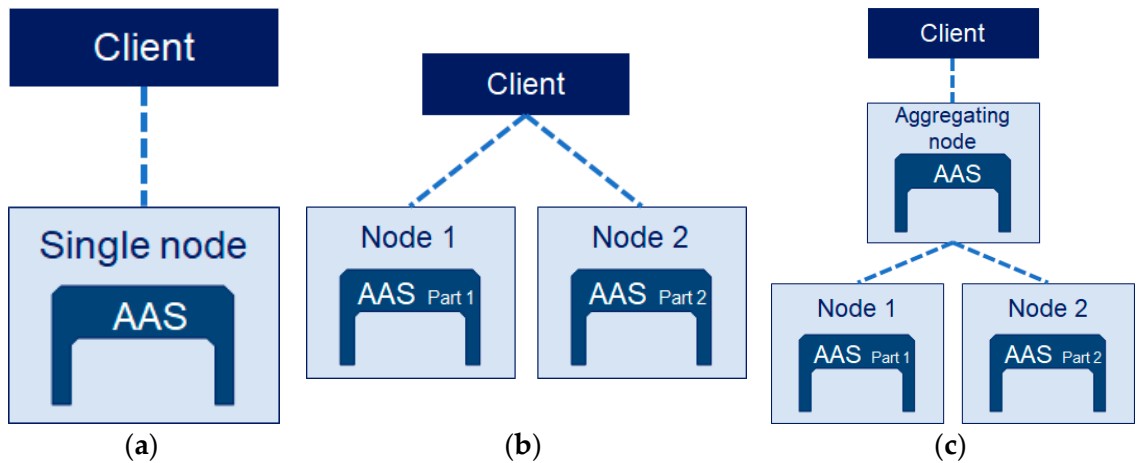

(**a**)  (**b**)  (**c**)

**Figure 3.** Distribution perspective for implementation of the AAS: (**a**) centralized, (**b**) distributed with loose-coupling, (**c**) distributed with aggregating node. Adapted from [42] with permission from the Federal Ministry for Economic Affairs and Climate Action. Scharnhorststraße 34–37, 10115 Berlin.

In Figure 2, the implementations differ in the distance between the AAS execution environment and the asset. However, no further consideration is made concerning the specific execution environment, which defines a form of AAS virtualization. This form of virtualization is an implementation perspective that implies advantages and disadvantages for the application. Three possibilities are presented and illustrated in Figure 4: the implementation based on Figure 4a operating system, Figure 4b hypervisor, and Figure 4c container. In the first case, the operating system is the AAS execution environment itself; that is, the execution environment consists of a process of a dedicated operating system or running inside another process. In the second case, the AAS execution environment is a virtual machine (VM). Multiple virtual machines with their own operating systems are allocated to a host (host) machine; they use its hardware and a hypervisor manages it. As in the previous case, AAS would still function as a complete operating system process but, in this case, this process would share hardware resources with other operating systems and applications. Finally, in the third possibility, the AAS execution environment consists of containers which run on top of an operating system. Unlike virtual machines, applications run in containers are run on the host machine's operating system, requiring only minimal resources such as applications and APIs needed to run the application, in this case, the AAS [43]. Two issues are related to the AAS execution environment, namely to its virtualization form: isolation and performance. These two characteristics form a trade-off, as pointed out by [44].

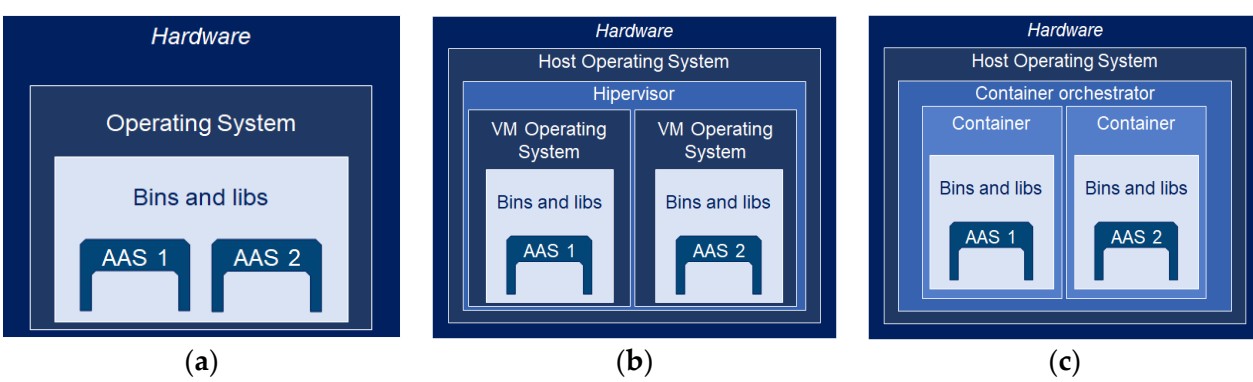

**(a)**           **(b)**           **(c)**

**Figure 4.** Virtualization perspective for implementation of the AAS: (**a**) operating system, (**b**) hypervisor, (**c**) container. Adapted from [42] with permission from the Federal Ministry for Economic Affairs and Climate Action. Scharnhorststraße 34–37, 10115 Berlin.

## 4. Results

This section presents results regarding the importance of technologies and methods gathered under the term "Big Data and Analytics" for I4.0 and correlates the characteristics of the data in this context with those of the relational and NoSQL databases.

### 4.1. Data Characteristics in the Context of I4.0

There is still no consensus on the technological pillars that support Industry 4.0 (I4.0). It is also observed that most authors put "Industry 4.0" and "Fourth Industrial Revolution" as synonyms, considering that there is no distinction between the phenomena, making it even more challenging to identify the technological pillars associated with each of the concepts. Table 1 presents the views of different authors about the key enabling technologies of the I4.0.

Despite this mentioned lack of consensus, it can be observed that there are certain convergences between authors and organizations about the enabling technologies of I4.0. It can be seen from Table 1 that the only technology identified as such in all the works consulted was big data and analytics. Thus, although there is no total consensus among the authors, the relevance of big data and analytics for the industry in the coming decades can be recognized. In brief, the term big data and analytics comprises (i) data sets "characterized

by a high volume, velocity and variety to require specific technology and analytical methods for its transformation into value" [45], as well as the technologies and data analysis methods (big data analytics) themselves for this type of information asset, that is, specific to its characteristics [46,47].

Data generated, collected, transmitted, and possibly analyzed in real-time will be part of the Smart Factory [48]. The fact that big data and analytics systems are considered a pillar of I4.0 comes from the possibilities of improvements that can occur in a company based on its available data. The authors of [17,48] state that this type of system can be used to increase productivity and for better risk management, cost reduction, and aid in general decision making. For this reason, this set of technologies is considered fundamental for I4.0 [49]. Considering that big data and analytics is a key enabling technology of I4.0, the five dimensions of big data presented can be used to obtain a more general description of the nature of data in the context of I4.0. Still, based on the definition of big data presented, four dimensions characterize this information asset: the volume, speed, variety, and value of data. A fifth dimension—veracity—is still considered by some authors to encompass the reliability of the data [48,50]. These five dimensions, also called "5V", characterize big data and directly affect how data is stored, manipulated, and analyzed in IMS [51]; that is, they require technological solutions and specific methods for these functions. A description of these five dimensions of big data is presented in the next subsection with a discussion of how they are manifested in I4.0, taking into account the concept, structure, and implementation perspectives of the Asset Administration Shell and their impact on the design of databases for Intelligent Manufacturing Systems.

**Table 1.** Key enabling technologies for Industry 4.0 and/or the Fourth Industrial Revolution, according to different authors.

| Technology | Rüssmann (2015) [52] | Bechtold et al. (2014) [53] | Lichtblau et al. (2015) [46] | Bauer et al. (2015) [54] | Petrillo et al. (2018) [55] | Wan, Cai, Zhou (2015) [56] |
|---|---|---|---|---|---|---|
| Big data and analytics | x | x | x | x | x | x |
| Advanced robotics | x | x | | x | x | |
| Systems integration | x | | | | x | x |
| Internet of things | x | | x | x | x | x |
| Simulation | x | | | | x | |
| Additive manufacturing | x | x | x | x | x | |
| Cloud computing | x | x | x | | x | x |
| Virtual/augmented reality | x | | x | x | x | |
| Cybersecurity | x | | | | x | |
| Machine-to-machine | | x | | x | | |
| Mobile technologies | | x | x | | | |
| Location and detection technologies | | | x | | | |
| Human–machine interfaces | | | x | x | | |
| Authentication and fraud detection | | | x | | | |
| Smart sensors | | | x | | | |
| Interaction with customers | | | x | | | |
| Community platforms | | x | | | | |
| Embedded projects | | | | | | x |
| Self-guided vehicles | | | | | | x |
| Social networks | | | | | | x |

### 4.2. Data Models in the Context of Industry 4.0

Section 3.2 allows for demonstrating that entities that seek to lead the evolution process of Industry 4.0 are clearly concerned with the description of data at a conceptual level. The creation of conceptual dictionaries, such as the aforementioned eCl@ss, IEC 61360, among

others, defines the composition and semantics of the elements that make up the AAS. In terms of databases, it can be seen that efforts are directed towards building conceptual data models for I4.0. The same concern is not observed regarding the mapping of conceptual models at the logical level. The systematic review results, whose procedure was detailed in Section 2, demonstrates the gap that exists in the proposals for system architectures for I4.0 in terms of database design. Of the 139 papers resulting from the application of the search string on the Google Scholar platform, which reports the implementation of the AAS, 25 of them scored higher than five based on the criteria adopted and were analyzed. Only 32% (8 papers) at least inform the data model, or DBMS used, suggesting that studies in this area discuss this issue

Based on the results obtained from the systematic literature review, it is possible to identify the lack of rigor in the choice of logical data models that are used in databases for AAS implementations. Alternatively, several works discuss the applicability of data models in scenarios that can be observed in Industry 4.0 but consider a specific application [57–59] or do not propose a direct correlation with the phenomenon and its particularities [60,61]. Despite this finding, databases should not be understood as mere tools for data storage but as essential components of architectures, impacting their performance [16]. For this reason, database solution choices should not be made arbitrarily but based on criteria, application characteristics, users, and data. Given the reality exposed in the systematic review, i.e., the gaps in the implementation of database solutions in the context of I4.0, a correlation between the data characteristics described in the previous section—considering the five dimensions of big data—and the characteristics of relational and NoSQL data models can be introduced, discussing the adequacy of these models to the context of I4.0.

### 4.2.1. Operational and Analytical Databases

The subsections dealing with the dimensions of big data and analytics and the Asset Administration Shell make it possible to have a characterization of the data in the context of Industry 4.0. However, these characteristics manifest differently depending on application specifics. Therefore, in order to have a characterization of the data that allows for a deeper discussion about the suitability of data models for Industry 4.0, a brief differentiation between two types of databases, operational and analytical, is proposed. They differ in terms of the type of operations they perform most frequently, the volatility of the data stored, the number and type of users, the volume of data, their generation, and processing speed, among other characteristics, which are briefly described as follows:

- Operations: Operational databases are generally dedicated to online transaction processing (OLTP) applications, routine operations of an organization, which act on small fractions of the data, occur with great frequency, and must be processed efficiently, usually in real time [25]. Such operations are, for example, insertions, updates, deletions, and queries [26]. Analytical databases, in turn, are optimized for online analytical processing (OLAP), which allow you to extract value from the data through complex analytics. They are mainly dedicated to data recovery, involving grouping and joining operators, statistical functions, and complex Boolean conditions [26], applied to a large number of records and which, for this reason, usually occur in batches;
- Volume and volatility: Data analysis requires not only a comprehensive system perspective but often a considerable amount of historical data, such as time series. Thus, an analytical DB stores volumes of data much larger than its data sources—transactional DBs—in addition to ensuring the persistence of this data for much longer, while traditional DBs usually store current and non-historical data [62]. Thus, changes in the content of an analytical DB usually occur incrementally and in batches, while changes in transactional DBs occur continuously [26,62];
- Orientation and users: Analytical DBs are systems dedicated to facilitating the exploratory analysis of data, aiding decision-making and business processes [24]. For this reason, they are said to be subject oriented and generally dedicated to few users. Operational DBs, in turn, are purpose oriented and may have few to many users [62].

4.2.2. Volume

Volume is associated with the amount of data involved in big data applications. Although there is no consensus on a reference value for data volume for a database to be characterized as big data, this dimension is usually characterized by petabyte (PB) or exabyte (EB). According to [52], data volume in the modern industrial sector tends to grow by more than 1000 EB per year. In the context of I4.0, the digitization of the most diverse assets and the communication between them leads to an "unprecedented growth" in the volume of data, according to [56]. The authors of [49] state that big data cannot be manipulated in a single computer, requiring a distributed architecture for IMS. Thus, it is noted that the concern with the volume of data in I4.0 is reflected in the perspectives of implementing the AAS concerning its distribution.

The need for database distribution is a characteristic that imposes limitations on using the relational model for large volumes of data [63,64]. The main problem associated with using relational databases to implement distributed database systems is linked to the CAP theorem and, consequently, to ACID transactional properties. A single server system is a CA system: there is no partition tolerance because there is no partition on a single machine. Therefore, the two other properties—availability and consistency—are guaranteed. Most relational database systems are CA, and licenses for this type of DBMS are generally marketed to run on a single server [27].

On distributed systems, there is the possibility of partitioning the network. In this type of system, it is only possible to leave off partitioning tolerance if, in the event of a network partition failure, the system becomes completely inoperative, which is critically undesirable in some instances. Thus, it is generally not desirable to leave off the tolerance of network partitioning; that is, it is not desirable to have a distributed CA system. The other possibilities are leaving off consistency or availability. Therefore, the essence of the CAP theorem can be understood as: in a system that may be subject to partitioning, one must prioritize between consistency or availability. This turns out to be, in fact, a trade-off between consistency and latency: to have consistent transactions, a certain amount of time is needed for data changes to propagate to all copies, and the system can be available again [33].

Because transactions that adopt the ACID model are strongly consistent, it is impossible to balance the trade-off between consistency and latency in distributed databases that use this model of transactional properties [33]. For this reason, maintaining ACID properties generally implies higher latency [27,65] in a distributed database that implements the relational model. For high availability, data needs to be replicated across one or more nodes, so if one node fails, the data is available on another. Replication can increase availability and performance by reducing the overhead on nodes for reading operations. However, for "write" operations, where one wants to ensure that all nodes have an up-to-date copy of the data, one can experience a loss of performance (one must wait until the data is replicated across all nodes). On the other hand, in systems that adopt the BASE model, lower latency can be achieved, but inconsistencies can occur during a specific time interval (inconsistency window) since the different nodes can present different versions of the same record. Thus, it is understood that NoSQL systems, especially those oriented to aggregates, are beneficial for I4.0 in distributed database implementation scenarios, which usually contain large volumes of data, as they facilitate horizontal scalability.

In addition to performance, aggregate orientation is another reason why NoSQL data models better suit distributed database systems. Specific applications may contain data sets that are frequently accessed and manipulated together. In distributed systems, these sets can form a natural distribution unit [27] so that interested users are always directed to one or more specific nodes where they are stored. In databases, these sets are called aggregates: a rich structure formed by a set of data (objects) that can be stored as a unit, as they are often manipulated in this way. Elements of AAS as Submodel (set of Submodel Elements), Submodel Element Collection, and AAS itself (set of other elements) can be

understood as aggregates. As such, aggregate-oriented NoSQL models are helpful for distributed implementations of AAS.

In terms of the implementation data model, the main problem of the relational model regarding the representation of aggregates is its rigid structure, which makes it impossible to treat data sets as a unit. The relational model allows representing the entities and relationships that are part of an aggregate. However, it does not allow to represent the aggregate itself; that is, it does not allow identifying which relationships constitute an aggregate nor the boundaries of this aggregate. For applications looking to process aggregates as a whole, the NoSQL key-value data model is beneficial because the aggregate is opaque. In case it is necessary to access parts of an aggregate, the document data model is more suitable than the key-value, as the aggregates are transparent in the case of the former; that is, partial operations can be performed on the data of the aggregates. For processing data from simpler formats such as numeric values, strings, etc., column family is also suitable for the purpose.

In summary, it can be argued that centralized implementations are suitable for smaller volumes of data, while distributed implementations are suitable for large volumes. There are relational databases that can be horizontally scalable; that is, they can be distributed [66] but the possible high unavailability of the system can make this distribution unfeasible. The trade-off between consistency and latency can be associated with two other dimensions of data—veracity and velocity—respectively, so that the speed dimension alone is not able to determine a more adequate data model.

### 4.2.3. Variety

Variety refers to the different formats of data. Big data applications can involve structured data, such as rigidly structured tables populated with scope-limited values; semi-structured as documents with a pre-defined template; and unstructured, such as multimedia content (image, audio) [67]. It is possible to argue that such heterogeneity may exist in the industrial context but it is possibly more "controlled". Despite this fact, variety is still a characteristic of the data in I4.0, considering that the AAS proposal presupposes a standardized format of representation and exchange that must be able to include assets of the most diverse natures. Thus, variety is a feature of the data in I4.0.

In the context of I4.0, this inability (or difficulty) can be verified in the attempt to create an AAS metamodel in a relational schema. Since the AAS must contemplate all I4.0 assets, it has a vast number of classes (entities) that represent each of its elements that would be translated into a large number of relations that could be even more significant if normalization procedures are applied. Associated with this complexity arising from mapping the AAS metamodel in the relational scheme, assets also have heterogeneity. When building a database composed of different assets, this heterogeneity can imply many null fields, which is undesirable.

Scenarios in which data heterogeneity is present may require flexibility in databases. A flexible data structure is not observed in the relational model, both in terms of the relationship scheme and the restrictions imposed by domains on possible values for attributes in the relationships. The flexibility of NoSQL systems makes them suitable for I4.0. Such flexibility enables the storage of semi-structured data, which best characterizes AAS. Technical reports from the Plattform Industrie 4.0 [34] and academic papers [68–70] present AAS implementations in XML and JSON format, which suggests that document-oriented NoSQL systems are advantageous, although it is not the only one capable of storing semi-structured data. This document encoding type is supported by essential communication technologies relevant to the I4.0, such as OPC UA [71] and HTTP. In summary, NoSQL data models adapt to the characteristic variety of data in I4.0, enabling the storage of heterogeneous records in the same DB. Thus, the flexibility of NoSQL models has its importance in the context of I4.0.

4.2.4. Velocity

This dimension can be understood as having two components—the velocity with which data is generated and the velocity with which it is processed [57]. In older big data applications, processing was commonly performed in batches, so the velocity at which data is generated and captured is critical to ensuring its reliability. Newer applications enable data processing in real time and in data streams so that, in addition to ensuring data reliability, the generation velocity must be consistent with the data processing velocity [52,54,57]. These two forms of processing are essential to guide the choice of database.

In addition to the high growth rate of data volume, which has already been mentioned and is more associated with data capture velocity, one can also discuss data processing speed and data analysis in IMS. Batch processing consists of processing a large volume of data at a time. The literature reports examples of this type of processing in an industrial environment [58,59]. Furthermore, Data Warehouse systems are typical examples of this way of processing and analysis. Applications of real-time processing and analysis in an industrial environment are also reported in the literature [60–62]. Comparing the AAS implementation platforms—edge, fog, and cloud—the last two are more suitable for batch processing since, in general, they have greater computational capacity than the first. However, this finding can be changed with the evolution of technology and the possibility of increasing data processing and storage capacity in devices closer and closer to the edges of networks. The asset-based implementation, that is, on edge devices, favors real-time processing due to low response latency.

Before introducing the discussion of data models suitable for batch and real-time processing, it is essential to introduce the Speed Consistency Volume principle, or SCV principle. While the CAP theorem presented in Section 3.1.2 concerns data storage, the SCV principle deals with data processing. The first attests that it is impossible to simultaneously guarantee consistency, availability, and tolerance to the partition. The second states that it is impossible to guarantee processing speed, consistency of results, and processing large volumes simultaneously. Based on [72], each of the elements that compose the SCV principle is described:

- Speed: deals with the speed at which data can be processed. To calculate the processing speed, the time spent to capture data should not be taken into account, considering only the actual processing time;
- Consistency: concerns the accuracy and precision of the results obtained from data processing. Inconsistent systems cannot use all available data to be processed, adopting sampling techniques, which leads to less precision and accuracy of results. On the other hand, systems with greater consistency use all available data in processing, obtaining more precise and accurate results;
- Volume: deals with the amount of data that can be processed. Large volumes of data require distributed processing, while smaller volumes can be processed centrally.

To analyze the "velocity" dimension, batch processing is initially considered. This type of processing is generally applied to analytical databases, which store large volumes of data and value the precision and accuracy of the results. Thus, from the point of view of the SCV principle, the properties that manifest themselves in this type of processing are volume and consistency. Analogously, considering the CAP theorem, the demands for consistency and tolerance to partition are manifested at the expense of speed. Thus, batch processing is often characterized by longer response times. Thus, data models that enable distribution and ensure data consistency are more suitable. Still regarding the velocity dimension in big data, the case of real-time processing is now analyzed. This type of processing is often used in operational databases. For those which store small volumes of data, there is not necessarily a distribution requirement, so the database system can be classified from the point of view of CAP theorem as a CA system, where the availability is low and consistency is ensured. In these cases, data models that implement ACID transactional properties are recommended. In cases where data have complex connections to each other, graph databases are particularly more efficient. For operational databases

with small data volumes, the implementation of AAS based on edge computing platform is suitable for this type of processing, as being closer (or even embedded) to the asset, delays tend to be smaller.

Operational databases can also contain large volumes of data that cannot be left off, which imposes the need for storage and processing distribution. Thus, there are trade-offs between processing speed and consistency of results (SCV principle) and between availability and consistency (CAP theorem). In real-time processing, as delays are un-wanted, trade-offs tend to prioritize speed and availability over consistency. It is known that ACID transactional properties do not allow the relaxation of consistency in favor of increased availability. Thus, aggregate-oriented NoSQL systems, which implement the BASE model of transactional properties, may be more suitable solutions for real-time pro-cessing. However, the level of consistency required by the application must be taken into account so that, by maximizing availability and processing speed, precision and accuracy requirements are not violated. Aggregate-oriented models are even more efficient; they guarantee higher processing speed if they do not need to perform operations on multiple aggregates simultaneously.

### 4.2.5. Veracity

Veracity is associated with the reliability of the captured data. The authors of [64] argue that the veracity dimension has three components: objectivity/subjectivity, which is more linked to the nature of the data source; deception, which refers to intentional errors in the content of the data or malicious modifications thereof; and implausibility (implausibility, irrationality) of the data, which refers to the quality of the data in terms of its validity, that is, its degree of confidence. Such concern is observed in the context of I4.0 as authors consider cybersecurity as a pillar of I4.0 (see Table 1), which presupposes protection against errors and intentional modifications to the data. It is also observed in the AAS implementation perspectives, where the virtualization strategy directly affects the isolation between applications [65] and, consequently, confidentiality and data integrity.

Some causes for veracity problems associated with implausibility, such as inconsis-tency, latency, and incompleteness, are pointed out by [67]. Therefore, it is observed that these causes and, consequently, the veracity is essential for the database design.

It is possible to associate the causes of implausibility with the properties of the CAP theorem and thus discuss the "truthfulness" dimension for different database systems. The inconsistency that affects the veracity of the data is directly linked to the consistency referred to in the CAP theorem. Latency is associated with the availability property, as seen in Section 3.1.2. The issue of incompleteness, in turn, is not directly associated with a property of the CAP theorem but with the transactional guarantee of atomicity, which establishes that a transaction must be performed entirely or not be performed at all. Thus, there is a foundation to discuss the impact of data models on veracity.

ACID transactional properties contribute to data veracity by enabling consistency and atomicity to operations. However, such properties imply high unavailability, which translates into delays in operations. Returning to the "speed" dimension of big data, if the data processing speed obtained through a system with ACID properties is consistent with the speed of data entry into the system, so there is no processing of outdated data, then these databases systems can be employed. Relational and graph-oriented DBMS generally adopt such properties.

Adopting the BASE model of transactional properties promotes an increase in avail-ability at the expense of relaxation of consistency, which implies a decrease in the delay but rises the possibility of occurrence of inconsistencies. This does not mean that the BASE model is a bad choice when one wants to guarantee veracity based on completeness and consistency. The BASE model does not make it impossible to guarantee consistency but allows a balance of the trade-off between consistency and availability to better suit the application's need. Thus, one of the properties can be prioritized according to the charac-teristics of the application and the problems related to implausibility, whose susceptibility

to the occurrence is greater. Thus, aggregate models can guarantee veracity, dealing with delay, incompleteness, and inconsistency, not simultaneously but balancing the problems according to the application demand.

This discussion is enriched with the distinction between analytical and operational databases. Analytical databases often have large volumes of data as they have less volatility. Therefore, they are generally implemented in distributed architectures, where the concurrency control problem is naturally less critical, especially when using aggregate-oriented data models, which allow users interested in a specific fraction of the data to always consult the network node that contains this fraction of the data, minimizing concurrent accesses. Additionally, analytical databases generally have fewer users, which further reduces the need for tight concurrency control. For these reasons, databases that implement the BASE model of transactional properties become a more suitable option. Operational databases have a greater number of users and, for this reason, they need to perform concurrency control more rigidly. For these cases, data models that implement the ACID model of transactional properties emerge as more viable options.

Finally, one can also consider in the discussion of veracity the differentiation between integration and application databases. The former store's data from multiple applications in a single database. This type of system has a much more complex structure than would be required by individual applications, as there is a need to coordinate and orchestrate applications, which differ above all in terms of performance requirements in terms of their operations. Application databases, in turn, are accessed and updated by a single application. This type of implementation allows databases to be encapsulated to applications, and the integration between them occurs through services so that application databases are fundamental for web applications and service-oriented architectures in general [27]. In an I4.0 context, it is possible to observe that the AAS implementation perspectives regarding its virtualization support both types of databases, especially concerning the degree of isolation between applications.

Integration databases generally implement the relational model, as the ACID properties precisely confer the desired concurrency control to coordinate the requests of different users/applications of the database [73]. However, for application databases, the relational model entails specific unnecessary and even undesirable characteristics: an application database usually requires a much smaller number of operations offered by the SQL language [27] and ACID transactional properties, which ensures concurrency control becomes unnecessary as only one application accesses the database [27].

### 4.2.6. Value

This dimension is associated with the value that can be extracted from the data through data analytics. Extracting value from data consists of converting the data into entities with a higher hierarchical level [57]. It involves a series of data analysis techniques, including machine learning, that requires a multidisciplinary approach, and, above all, it receives the name "value" because it offers prospects for improvement and cost reduction in terms of products and processes in the industry [52,63] so that it is possible to argue that there is a loss in not extracting value from the data. Extracting value from data in a significant data context presupposes the application of specific technologies and analytical methods. This dimension highlights the importance of data for the industrial sector as it brings the possibility of implementing improvements in organizations based on data.

Extracting value from data involves employing data analysis techniques in a multidisciplinary approach, which can translate into a naturally distributed organizational structure of a company or institution. Regarding the AAS implementation perspectives, its distribution in different network nodes, in fact, better reflects the structure of organizations today [41]. In these situations, an organization's subdivision is responsible for a fraction of the AAS or the whole AAS that concerns it. Because applying data analysis techniques in an organization to extract value from them requires the integration of different perspectives, it assumes that distributing AAS across different nodes also requires that these "AAS

fragments" (or different AAS) be integrated to extract value from its data. Taking up the perspectives of AAS implementation regarding its forms of distribution, the distributed solution with the aggregator node can be an adequate solution for data integration. This does not mean that the aggregator node needs to act as a master node of the network, which manages all routine transactions, but that it can act as a data integration node and as a source of access by those members of the organization who intend to extract data value.

Although the extraction of value from the data is more linked to data analysis than to its storage, here is an excerpt of this dimension regarding database management systems, considering the interdisciplinarity and distribution of data in organizations. Thus, the analysis of the adequacy of database systems for this "value" perspective is mainly achieved by taking into account the importance of AAS distribution and the need for integration to extract value from asset data.

Network nodes that only contain AAS data referring to an organizational unit of the institution are those that process more routine transactions, the so-called online transaction processing. The databases of these nodes are called operational or transactional databases. A network node that promotes data integration, in turn, processes transactions with an analytical purpose, online analytical processing, and provides data for algorithms and other subsystems, acting, in fact, as an integrator database. In an institution with a distributed organizational structure, the data models that enable the horizontal scalability of the database, that is, the NoSQL DBMSs oriented to aggregates, are suitable for implementing "transactional" nodes, that is, those that process routine data transactions' specific AAS that pertain to a unit of the organization. If AAS distribution is not performed through the DBMS itself but through application databases that communicate by means of service interfaces, then NoSQL data models are still applicable. The flexibility provided by these models allows each company's subdivision to adopt the data models that best suit their application.

An integrator node is usually built from the so-called multidimensional data model at the conceptual level of data abstraction [74,75]. This is where the value is extracted from the data utilizing (big) data analytics methods. The multidimensional model is generally mapped at the implementation level through a relational scheme, although there are literature works that seek to map the multidimensional model in NoSQL models [76–78]. In particular, the importance of this mapping for the graph model is highlighted: the integrator node usually performs processing in batches and, based on the discussions in the previous subsection, it was argued that the graph-oriented model is suitable for this type of processing.

## 5. Discussion

Big data dimensions and other data characteristics in I4.0 were addressed in the previous subsections to discuss the suitability of data models to different realities of I4.0. However, interrelationships among these characteristics are analyzed for the database design, as it is possible to observe that one dimension can affect the others regarding the data model to be used. The dimensions "volume" and "velocity", for example, are correlated according to the SCV principle. When dealing with the "veracity" dimension, the impact of the BASE and ACID models on the veracity of the data was discussed. However, using one or another model of transactional properties also affects the distribution of data associated with the "value" dimension. The variety dimension, which concerns the possibility of storing data with more complex structures and, therefore, presupposes the use of more flexible data models such as those oriented to aggregates, for example, also implies the speed of processing multiple records, which is inferior in this type of data model.

Two qualitative analyses are presented to synthesize the results of the last section. The first of them is represented in Table 2, in which the dimensions "volume", "velocity", and "veracity" are associated with the two models of transactional properties, that is, BASE and ACID. As seen earlier, the first model is generally implemented in aggregate-oriented NoSQL databases, while the second is implemented in relational and graph databases.

Table 2 also includes the type of database—analytical or operational—where the scenario is more likely to be observed.

**Table 2.** Most suitable model of transactional properties according to the volume, velocity, and veracity of data.

| Volume | Velocity | Veracity | Database Type | Suitable Model of Transactional Properties |
|--------|----------|----------|---------------|--------------------------------------------|
| Low | Low | Low | Operational | BASE |
| | | High | Operational | Both |
| | High | Low | Operational | BASE |
| | | High | Operational | ACID |
| High | Low | Low | Analytical | BASE |
| | | High | Analytical | Both |
| | High | Low | Operational | BASE |
| | | High | Analytical | BASE |

The recommendations for each of the lines of Table 2 are presented here.

- The first scenario characterized by low volume, velocity, and veracity is more likely to be observed in operational DBs even though low velocity is usually not desirable. Considering the correlation between veracity and consistency presented in Section 4.2.5, the BASE model allows flexibility of consistency and, consequently, of veracity. This is the determinant factor for this recommendation;
- The second scenario with low volume and high veracity is again more likely to be observed in operational DBs despite the low velocity. As there is no need for distribution due to the low volume nor high-velocity requirements, ACID and BASE can ensure high veracity;
- The third scenario with low volume, high velocity, and low veracity can represent an operational DB. As there is a demand for high velocity at the expense of veracity, the BASE model is more suitable as it allows for relaxation of consistency in favor of availability;
- The scenario with low volume and high velocity and veracity well represent an operational DB as well as an analytical DB in its early stages. Since the database design needs to take into account the evolution of the DB, this analysis is made considering the former. Based on the CAP theorem and the SCV principle, the requirements of high speed and veracity imply the need for centralization of the database so that it is not subject to partition. Since the volume of data considered is small, there is no problem regarding distribution. For a CA system such as this, the ACID model is more suitable;
- Despite the low veracity, the fifth scenario may better represent an analytical DB than an operational DB. Although this type of DB requires high veracity (consistency), the distribution and the lack of strict concurrency control make the BASE model more suitable;
- The sixth scenario represents an analytical DB well. The requirement of high veracity (consistency) at the expense of speed can be initially associated with the ACID model. However, the distribution and no need for strict concurrency control present in an analytical DB mean that the BASE model can also be used;
- The scenario with high volume and velocity and low veracity illustrates an operational DB well. As there is a demand for high speed at the expense of veracity, the BASE model is more suitable as it allows for relaxation of consistency in favor of availability, especially in a distributed system;
- Regarding the eighth scenario, even though it is ideal for both an operational and an analytical DB, based on the CAP theorem and the SCV principle, it is not possible

to guarantee the three properties simultaneously neither with the BASE model nor with ACID. However, it is important to recognize that the very nature of the analytical distributed DB without the need for concurrency control contributes to high veracity. Thus, in an analytical database, to ensure distribution of a large volume of data and high processing speed, the BASE model can be used;

Table 2 does not refer to one or more data models specific to each scenario. The "variety" dimension, in addition to data linkage complexity and the flexibility of access, can be taken into account so that, based on a model of transactional properties, the choice for a data model can be made. Table 3, inspired by [51], synthesizes these three characteristics also qualitatively, suggesting the most appropriate data model with ACID transactional properties. Likewise, Table 4 suggests the most suitable data models with BASE transactional properties according to the veracity dimension, access flexibility, and data linkage complexity.

**Table 3.** Most suitable data model with ACID properties according to veracity, access flexibility, and data linkage complexity.

| Variety | Access Flexibility | Data Linkage Complexity | Suitable Logical Data Model |
|---------|---------|---------|---------|
| Low | Low | Low | Relational |
| | | High | Graph |
| | High | Low | Relational |
| | | High | Graph |
| High | Low | Low | Graph |
| | | High | Graph |
| | High | Low | Graph |
| | | High | Graph |

**Table 4.** Most suitable data model with BASE properties according to veracity, access flexibility, and data linkage complexity.

| Variety | Access Flexibility | Data Linkage Complexity | Suitable Logical Data Model |
|---------|---------|---------|---------|
| Low | Low | Low | Key-value |
| | | High | Document |
| | High | Low | Column family |
| | | High | Document |
| High | Low | Low | Column family |
| | | High | Document |
| | High | Low | Column family |
| | | High | Document |

Initially, database recommendations that implement the ACID model of transactional properties are discussed. For scenarios where data have high complexity in connections, the graph model is strongly recommended. This type of data model is also ideal in scenarios with high variety, where the rigid structure of the relational model is a disadvantage. Both allow flexible access to data and, therefore, in scenarios with less variety and complexity of connections, the relational model emerges as a viable option.

Some considerations about the recommendations of data models which implement the BASE transactional properties are presented: Key-value databases can easily store data with high complexity and variety but, in these cases, the complexity of handling the data is transferred to the application that deals with the data since the aggregate is opaque. That is the reason why the key-value data model is only recommended here for the scenario with low variety and linkage complexity.

Column family databases are strongly recommended for analytical databases. The way related data are organized in groups (the column families) optimizes not only operations for retrieving records (especially for similar data), as the rows are indexed, but also aggregate functions such as statistical operations, as the column families are also associated with primary keys. This data model can provide high access flexibility, but the structure of the database needs to be previously known.

Document databases are strongly recommended for storing and handling unstructured and semi-structured data. That is why they are suggested over column family databases for scenarios with high variety and data linkage complexity. They also provide higher access flexibility in comparison to column family as the aggregate is transparent; metadata is encapsulated into the document.

It is possible to observe that, when considering the specifications of a given application along the dimensions, the choice for a database generates trade-offs in terms of the requirements that can be met. In specific applications, conflicting characteristics from a database standpoint can be equally important. For this reason, it is common to find applications, especially in service-oriented architectures, in which multiple databases are used to meet the different application specifications satisfactorily. Works in this area are referred to as polyglot persistence [16,27], in which each database is responsible for managing data from a part of the application.

Finally, it is important to highlight that there are other factors linked to the specific characteristics of applications that can significantly affect the performance of databases. Optimization solutions are also constantly being developed [79,80]. Prominence is given to the class of databases called NewSQL, which seek to optimize the scalability of traditional relational databases such as that of NoSQL systems. The mapping of the conceptual model to the logical model itself can impact the performance of the database, as pointed out in [81]. All these factors may eventually modify the recommendations presented in this work.

## 6. Conclusions

This work presents different contributions regarding the database in the context of Industry 4.0 (I4.0). Given the importance of understanding the characteristics of the data for the design of a database, this article provided a comprehensive description of the data in the present context, identifying, for this purpose, the technologies, methods, and standards related to data that show fundamentals for the I4.0.

Systems architectures that organize the fundamental technologies and methods to provide functionalities of Intelligent Manufacturing Systems (IMS) can be provided. An indispensable element for these architectures is the database. Regarding the design of this element, this paper seeks to corroborate the assertion that, among the works that propose architectures for IMS, including adopting its standardizations such as Asset Administration Shell (AAS), few demonstrate evident concern and justification about the choice of data models to be used and how databases can influence the performance of these architectures. Subsequently, based on the characterization of the data in an I4.0 context, analysis was made of how the characteristics of relational and NoSQL data models fit into the dimensions of the data—volume, velocity, variety, veracity, and value. These analyses were summarized in Tables 2–4, in which hypothetical scenarios were built based on four of the five dimensions of data and other characteristics such as flexibility of access and complexity of data connections. The transactional guarantee models (ACID and BASE) and the data models (relational and NoSQL) that best fit each scenario were suggested.

The results presented in this paper adopted a qualitative comparison between data models. Works found in the literature propose comparisons between the performance of relational and NoSQL databases based on quantitative metrics [32,57,82]. However, the dimensions dealt with in this article can be analyzed quantitatively. The velocity dimension is widely used for performance comparisons across databases. This dimension can be measured in terms of the time it takes for database instantiation, reading, writing, removing, and searching operations to be performed on the database, as conducted by [83]. The

volume dimension, strongly associated with the distributed databases, can also be evaluated quantitatively. In [84], the performance of databases is compared in terms of the number of operations performed per second, but having as parameters in the comparisons the amount of data stored and the number of nodes in the cluster where the data is distributed. Quantitative metrics for evaluating flexibility are presented in [85]. Data linkage and structure complexity can be quantitatively accessed by the metrics defined in [86]. Thus, future work can explore the dimensions by which the data were characterized in this paper and quantitatively assess the performance of the data models for the scenarios presented.

Furthermore, the previous section briefly mentions the concept of polyglot persistence, in which multiple databases are used in the architecture for the IMS as a whole or its subsystems. This work considered the use of a single relational or NoSQL data model for each scenario and then it was pointed out which would be a possible, most adequate choice. Future work can explore the combination of different data models for each scenario and discuss the possible improvements this combination would have, as well as the cost of managing more than one database for each application.

**Author Contributions:** Conceptualization, V.F.d.O. and F.J.; data curation, V.F.d.O. and F.J.; formal analysis, V.F.d.O. and P.E.M.; funding acquisition, F.J., M.A.d.O.P. and P.E.M.; investigation, V.F.d.O. and F.J.; methodology, V.F.d.O., F.J. and P.E.M.; project administration, V.F.d.O., F.J., M.A.d.O.P. and P.E.M.; resources, V.F.d.O., F.J., M.A.d.O.P. and P.E.M.; software, V.F.d.O.; supervision, F.J., M.A.d.O.P. and P.E.M.; validation, V.F.d.O., F.J., M.A.d.O.P. and P.E.M.; visualization, V.F.d.O., F.J., M.A.d.O.P. and P.E.M.; writing—original draft preparation, V.F.d.O. and F.J.; writing—review and editing, V.F.d.O., F.J., M.A.d.O.P. and P.E.M. All authors have read and agreed to the published version of the manuscript.

**Funding:** This research was supported by the Coordenação de Aperfeiçoamento de Pessoal de Nível Superior (CAPES), grant number 88887.508600/2020-00; Fundação de Amparo à Pesquisa do Estado de São Paulo (FAPESP, São Paulo Research Foundation), grant number 2020/09850-0; and Conselho Nacional de Desenvolvimento Científico e Tecnológico (CNPq. National Council for Scientific and Technological Development), grant numbers 303210/2017-6 and 431170/2018-5.

**Institutional Review Board Statement:** Not applicable.

**Informed Consent Statement:** Not applicable.

**Data Availability Statement:** The database from the systematic literature review is available upon request from the corresponding author of this paper.

**Conflicts of Interest:** The authors declare no conflict of interest. The funders had no role in the design of the study; in the collection, analyses, or interpretation of data; in the writing of the manuscript; or in the decision to publish the results.

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
