# Peer review of "SQL and NoSQL Databases in the Context of Industry 4.0"

_machines, doi:10.3390/machines10010020_

Round 1

Reviewer 1 Report

The contribution of this paper is to propose suitable DBMS for the problem of AAS in the Industry 4.0 context.
The chosen DBMS classes are Relational and NoSQL-like.

The tackled problem is essential to Industry 4.0 because there are many classes of DBMS, and choosing one is a challenge.

In the following, I point out my concerns:
- The paper is probably more verbose than necessary.  Maybe through careful editing and brief discussion can be cut by 20% with no loss of information content. For example, the 5 V's of big data are presented twice (Sections 4.1 and 4.2). BTW, velocity: the speed at which data are GENERATED and processed. I think "collected" is not the correct name.
- The authors should use relational instead of SQL. SQL is the language to deal with relational data
- I miss a related work section. I think other works are trying to solve the same problem
- In the Methods: How many papers are returned from query string? (the answer appears in Section 4.2 - wrong place). Moreover, the ranking should be better explained. Maybe, an example would help.
- Figure 1 seems useless (Figure 2 could be used instead)
- The presentation of AAS components metamodel is not suitable. The authors should have used UML class diagrams instead of EER. The former makes the class relationship easier to understand. 
- Is it possible to present an instance of AAS metamodel? (e.g., see Figure 8  from Digital Twin and Asset Administration Shell Concepts and Application in the Industrial Internet and Industrie 4.0 - www.iiconsortium.org)
- Figure 3 is lost in the text. It is referenced but with no detailed context. When the reader thinks it is going to be explained, the authors explain Figure 4.
- I'd like to see a deeper discussion of Tables 2, 3, and 4. 
--- The choices made. 
--- Why value is not considered. 
--- Why veracity is associated with transactions (BASE and ACID).
--- The reasons to choose a given model to be used in a given context. For example, Variety (Low), Access (Low) Data Linkage (Low) -> Key-value model (Table 4). The key-value model was chosen because it is the simplest one? I think the inspiration from [55] should be presented in the discussion.

Minors:
- Reference 57 is incomplete
- 618 (see Table 2) --> (see Table 1) ?
- 893 Section -> section
- 806 as seen in 3.1.2 --> as seen in Section 3.1.2
- I think NewSQL is missing in the discussion (maybe in the future work)

Reviewer 2 Report

This paper studies the advantages and disadvantages
of relational (SQL) and non-relational (NoSQL) data
models for Industry 4.0 (I4.0) applications, and
describes the suitability of SQL and NoSQL databases
for different scenarios in the context of I4.0.

The presentation of the paper is good. There are
sufficient details, background information, and
analysis of the related work. It is an excellent
survey for database solutions in the context of I4.0.

The study is also focused on big-data 5V (volume,
variety, velocity, veracity, value), and proposes
the most suitable model for transactional, ACID 
and BASE properties.

Author Response

Reviewer 2 did not indicate possible points for improvement in the article. We are grateful for the considerations made.

Reviewer 3 Report

This paper provides a state of the art in NoSQL database systems and proposes an analysis NoSQL database in industry context.

The paper is interesting and focus on a real problem. 

The paper introduces some preliminary concepts and description of NoSQL systems and lists some characteristics that are
explained, motivated, and discussed.   

Section 3

The related work section could be improved. 

I would analyze better how the characteristics the variables taken into consideration in the classification are considered in other papers. 

Some NoSQL references are missing as:

Paolo Atzeni, Francesca Bugiotti, Luca Cabibbo, Riccardo Torlone:
Data modeling in the NoSQL world. Comput. Stand. Interfaces 67 (2020)
Z. Tan, S. Babu:
Tempo: Robust and Self-Tuning Resource Management in Multi-tenant Parallel Databases. Proc. VLDB Endow. (2016)

I cannot understand how Figure 1 us useful.

Author Response

(The authors gave the same response as above.)
